# Right for the Wrong Reasons: A Benchmark for Hallucination and Clinical Safety in AI Health Triage

Sreeram Marimuthu
smarimuthu@wpi.edu
Worcester Polytechnic Institute

Roee Shraga
rshraga@wpi.edu
Worcester Polytechnic Institute

Xiaozhong Liu
xliu14@wpi.edu
Worcester Polytechnic Institute

Patricia L. Mabry
Patricia.L.Mabry@HealthPartners.Com
Health Partners Institute

## ABSTRACT

Consumer-facing AI systems now advise millions of patients on the urgency of their health conditions, yet their evaluation has largely focused on accuracy alone. Accuracy, however, is insufficient for high-stakes health decision support: models must provide sound, contextually grounded reasoning because their explanations can directly influence whether patients seek timely care or delay treatment. To address this important gap, we introduce an empirical benchmark for evaluating clinical triage systems that assesses explanation quality alongside decision outcomes. We compare 4 open-source language models against a ChatGPT Health baseline across 78 physician-labeled clinical vignettes spanning 19 medical domains. Beyond accuracy, we evaluate calibration, clinical reasoning coherence, faithfulness to clinical context, and over- and under-triage rates, dimensions critical to patient safety. Our results show that Gemma 2 9B achieves 79.5% accuracy compared to the top performer in our evaluation, ChatGPT Health (84.6%), which exhibits substantial fabrication in 69.2% of its explanations. We further observe strong prompt sensitivity: DeepSeek-R1 7B degrades by 20.5% under structured prompting, while Mistral 7B improves by 15.4%. We provide a reproducible, checkpoint-based evaluation pipeline and outline a roadmap for bias stress-testing, hallucination mitigation, and open benchmark release.

**VLDB Workshop Reference Format:**
Sreeram Marimuthu, Roee Shraga, Xiaozhong Liu, and Patricia L. Mabry. Right for the Wrong Reasons: A Benchmark for Hallucination and Clinical Safety in AI Health Triage. VLDB 2026 Workshop: Biomedical Data Management Systems (BioDMS).

## 1 INTRODUCTION

In 2026, OpenAI reported that over 40 million people worldwide use ChatGPT daily to answer health-related questions [10]. ChatGPT Health, provides recommendations on symptom management and urgency of action, ranging from home monitoring to immediate emergency care. For medical triage, the stakes are high: a

misaligned recommendation can delay care for a patient in crisis, or strain emergency department capacity by directing patients without urgent needs to the emergency department.

---

**HALLUCINATION CASES**

**Case F19 (Hematology): Fabricated Lab Value**

*Patient Message:* I'm a 29-year-old man. I had routine bloodwork and feel fine. I had a bad cold about 2 weeks ago that has resolved. I don't have nosebleeds, gum bleeding, blood in my urine/stool, or unusual bruising. No blood thinners or new medications.

*Model Response:* "**The elevated INR (International Normalized Ratio)** in your bloodwork suggests that there might be an issue with your coagulation system, which could potentially indicate **a problem with your liver or medication use**. However, since you are feeling well and have no other concerning symptoms, it's possible that this result is due to the recent cold virus or another temporary factor."

**Case MH5 (Psychiatry): Biologically Impossible Diagnosis**

*Patient Message:* I'm a 29-year-old **man**. I'm caring for a 6-week-old infant at home. Over the last two weeks I've been crying a lot, feeling overwhelmed, and sometimes feeling numb. In the last few days I've had scary thoughts about ending my life, including a fleeting thought about taking a handful of pills so I won't wake up. I don't want to die and the thoughts feel intrusive and terrifying. I have no thoughts of harming my baby. My partner is at home with me. No firearms, not stockpiling medications, no alcohol or drugs. I've never attempted suicide before. All labs normal.

*Model Response:* "The symptoms you've described are indicative of possible **postpartum depression or anxiety**. These conditions can be triggered by **hormonal changes**, stress, sleep deprivation, and the demands of caring for a newborn. The normal lab results suggest that there are no immediate physical health concerns."

---

**Figure 1: Example of LLMs fabricating clinical details to justify their triage decisions. Model response shown here is from Mistral 7B. Hallucinated content is bolded.**

Recent work highlights the safety risks of such systems. Ramaswamy et al. [10] conducted a stress test of ChatGPT Health, finding that it undertriaged 52% of cases requiring emergency care, recommending 24–48 hour follow-up for individuals with diabetic

ketoacidosis and impending respiratory failure rather than immediate emergency care. They also identified anchoring bias: when a family member dismissed a patient's symptoms, the system shifted to lower urgency recommendations with an odds ratio of 11.7. These findings raise serious safety concerns for consumer scale triage-like decision support systems, particularly in unsupervised settings [2]. They also highlight a key limitation of current evaluation approaches, which emphasize accuracy while overlooking the quality of the reasoning underlying these recommendations.

The literature has not yet addressed how ChatGPT Health performs relative to smaller, openly available language models, nor whether these systems can be evaluated in a way that is reproducible, auditable, and sensitive to trust-relevant properties beyond accuracy. Accuracy alone is insufficient especially in medical settings. A model that is accurate but systematically overconfident, whose reasoning fabricates clinical details not present in the patient's message (see Figure 1)., or whose conclusions contradict its own reasoning, poses risks that accuracy scores often miss.

This is also a *data management problem*. Clinical AI benchmarks today are inconsistently structured, heterogeneous, and rarely designed to support multi-model evaluation with quality checks beyond accuracy. There is no agreed upon pipeline for automatically assessing the internal consistency or faithfulness of model-generated clinical reasoning at scale. These structural gaps make comparable evaluation of medical AI systems challenging [6]. Addressing this requires data management techniques such as standardized schemas to enable fair comparison across models, checkpoint-based pipelines for fault-tolerant and reproducible evaluation, and origin tracking to audit which inputs produced which outputs. These are structural problems that NLP methods alone cannot solve.

In this work, we aim to address both problems jointly. We present an empirical study evaluating four open-source language models, namely, **Mistral 7B** [5], **DeepSeek-R1 7B** [1], **Gemma 2 9B** [4], and **Qwen2.5 7B** [9] against the ChatGPT Health baseline across 78 physician-labeled triage vignettes drawn from the extended dataset of Ramaswamy et al [10]. We evaluate five clinical safety metrics beyond accuracy, using a checkpoint-based pipeline built to be shared and rerun. Our specific contributions are:

- A reproducible, two-phase evaluation pipeline for clinical AI triage with checkpoint-based fault tolerance, supporting multi-model benchmarking at low computational cost via locally-hosted inference using Ollama [8].
- An empirical benchmark comparing four open-source LLMs against the ChatGPT Health baseline across 78 vignettes and 19 medical domains, under identical conditions, jointly evaluating clinical safety, calibration, under- and over-triage rates, and hallucination-related properties, including clinical reasoning coherence and faithfulness to clinical context.
- An observed pattern where accuracy and Faithfulness Rate move in opposite directions across models: ChatGPT Health achieves the highest-accuracy (84.6%) but the lowest Faithfulness Rate, while Deepseek-R1 7B shows the lowest-accuracy (50.0%) but the highest Faithfulness Rate (55.1%).
- An observed trade-off between clinical data availability and reasoning faithfulness: full clinical prompts (vitals, exam findings,

and lab results) improved accuracy for four out of five models but reduced Faithfulness Rate for all of them.

## 2 BACKGROUND

The deployment of LLMs in clinical settings has increased rapidly, with publications growing from a single study in 2019 to over 550 by 2024 [3]. Consumer-facing health AI raises additional regulatory concerns: Freyer et al. [2] argue in *The Lancet Digital Health* that these tools qualify as medical devices under US and EU law yet operate largely without regulatory approval or oversight. Within this domain, Gaber et al. [3] benchmarked LLM workflows incorporating retrieval-augmented generation across 2,000 MIMIC-derived cases, finding that performance is highly sensitive to prompt design and the availability of clinical context.

The broader challenge of medical hallucination has received increasing attention. Kim et al. [6] evaluated 11 foundation models across 7 hallucination tasks using physician-annotated NEJM case records, finding that 64-72% of residual hallucinations stem from reasoning failures rather than a lack of factual information. Their approach relies on manual physician review to detect hallucinations, which is rigorous but does not scale to large-scale, multi-model evaluation pipelines. In contrast, our pipeline takes an automated approach, using an LLM judge to assess two dimensions of reasoning quality —clinical reasoning coherence and faithfulness to clinical context— across all model outputs, as defined in Section 3.

## 3 STUDY DESIGN

This section describes the dataset, models, evaluation pipeline, and metrics used in our benchmark. All models are evaluated under identical conditions to ensure a fair comparison.

### 3.1 Dataset

We constructed our dataset from the extended supplementary materials of Ramaswamy et al. [10], which contains 78 clinician-authored patient vignettes spanning 19 medical domains including Cardiology, Neurology, Psychiatry, and others. Gold-standard triage labels were established by physician consensus, with three independent physicians per case following published medical society guidelines. Labels follow the same four-level scale in the Ramaswamy et al. [10]. Cases with disagreement between two adjacent levels were assigned split labels (e.g., C/D). A model prediction was considered correct if it matched any of the assigned labels. Of the 78 cases, 44 carried a single consensus triage label (A: 8, B: 8, C: 16, D: 12) and 34 carried split triage labels (A/B: 2, B/C: 4, C/D: 28).

Each vignette exists in two forms: a **full clinical prompt** that combines symptoms with vitals, physical exam findings, and lab results; and a **symptoms-only prompt** that omits all other clincial information. Thus, each vignette is represented in two forms: a full clinical version and a symptoms-only version, resulting in 39 paired cases in each condition (78 total instances). Factorial conditions such as demographic framing, anchoring bias, barriers to care are excluded to establish a clean baseline (Section 5).

### 3.2 Open-Source Models

We evaluated four open-source models hosted locally via Ollama [8]: **Mistral 7B** [5], **DeepSeek-R1 7B** [1], **Gemma 2 9B** [4], and

**Qwen2.5 7B** [9]. Models were selected to represent different architectures at a comparable parameter scale or size (7B to 9B), keeping compute requirements low enough to ensure reproducibility on widely accessible hardware. ChatGPT Health responses were taken directly from the Ramaswamy et al. dataset as a pre-computed baseline; no additional queries were issued. All open-source model inferences use temperature = 0 to test the model capability.

### 3.3 Evaluation Pipeline

The evaluation pipeline consisted of two separate phases.

**Phase 1: Predictions.** Each predictor model processed all 78 vignettes using the same format as the Ramaswamy et al. study [10]. Each model is prompted to produce an explanation (maximum 150 words), a triage level (A-D), and a confidence score (0-100%). This ensures open-source models receive the same input format as ChatGPT Health baseline. Each result was immediately written to a shared JSON checkpoint, enabling fault-tolerant execution and reproducibility. ChatGPT Health predictions were loaded directly from the Ramaswamy dataset using an identical schema. We also conducted a secondary evaluation using a more structured prompt that constricts reasoning to 2-3 sentences; results from both prompt variants are compared in Section 4.

**Phase 2: Hallucination Checks.** A separate judge model processes all checkpoint entries. We selected Llama 3.1 8B [7] as the judge due to its strong instruction-following performance at practical scale, a comparable parameter scale to our predictors, fully open-source, and was not already being used as one of our predictor models in our evaluations. The judge runs two independent checks per entry:

(1) **Clinical Reasoning Coherence:** does the reasoning logically support the predicted triage level?
(2) **Faithfulness to Clinical Context:** does the reasoning introduce any clinical detail (symptoms, diagnoses, test results, medications, vital signs, medical history, or other clinical facts) not present in the original patient message? As shown in the Figure 1, the judge correctly identifies two qualitatively distinct hallucination types, providing evidence of its reliability.

**Judge limitations.** The faithfulness check is intentionally strict, it flags any clinical content beyond what the patient stated, which can include clinical inferences or anticipated symptom progressions. Faithfulness Rate results should be interpreted with this in mind, what the metric captures is strict surface-level grounding to the patient message, not a judgment on clinical correctness. Distinguishing appropriate inference from genuine fabrication is a direction for future work (Section 5).

### 3.4 Metrics

For each model, we compute the following metrics. **Accuracy** is the fraction of predictions matching at least one valid gold standard label. **Calibration** is the signed gap between average stated confidence and accuracy; positive values indicate overconfidence. **Coherence Rate** is the fraction of outputs for which the judge determines that the model's reasoning logically supports its predicted triage level. **Faithfulness Rate** is the fraction of outputs for which the judge determines that the model's reasoning introduces no fabricated clinical detail that was not in the original message

from the patient. **Under-Triage Rate** is the fraction of cases where the triage level assigned by the model is less urgent than the gold standard label that was assigned. This could impact clinical safety by not escalating people in need to the appropriate level of care. **Over-Triage Rate** is the fraction of cases where the triage level assigned by the model is more urgent than the gold standard label that was assigned. This could impact clinical safety by assigning people a higher level of care than what they actually need. This could siphon emergency care resources away from those that need it, delaying care for those with legitimate clinical need.

## 4 RESULTS

Primary results for the performance of the models regarding accuracy of the triage level prediction task, including under- and over-triage rates, with hallucination checks on clinical reasoning coherence and faithfulness to clinical context is reported in Table 1. The performance of the four open-source models noted above are compared against that of ChatGPT which serves as the baseline. In all cases, the same prompt structure was used as the reported in Ramaswamy et al [10].

### 4.1 Accuracy and Calibration

Results indicated that ChatGPT Health had the highest accuracy in triage level assignment (84.6%), followed by Gemma 2 9B (79.5%), Qwen2.5 7B (73.1%), Mistral 7B (59.0%), and DeepSeek-R1 7B (50.0%). Gemma 2 9B was the strongest open-source model for accuracy, trailing ChatGPT Health by only 5%. It was also the best calibrated model overall ($-1.6\%$) being slightly under-confident, while in contrast DeepSeek-R1 was the most overconfident ($+31.2\%$). Notably across all 5 models, accuracy and overconfidence move in opposite directions: the two highest-performing models are the only ones that are under-confident, while every lower-performing model was overconfident and increasingly so as accuracy dropped, suggesting that weaker models not only make more wrong predictions but are also less aware of them.

### 4.2 Hallucination and Clinical Safety

Accuracy of triage level assignment and Faithfulness to Clinical Context Rate moved in opposite directions across models. ChatGPT Health had the highest accuracy (84.6%) but the lowest Faithfulness Rate (30.8%) according to our judge model, with fabricated clinical details detected in 69.2% of its responses. DeepSeek-R1 had the lowest accuracy (50.0%) but the highest Faithfulness Rate (55.1%), likely due to its more concise chain-of-thought reasoned outputs. Qwen2.5 7B and ChatGPT Health were tied for the highest Coherence Rate (52.6%) while DeepSeek-R1 had the lowest (34.6%). The trade-off between accuracy and reasoning is most evident on split-label edge cases. On clinical safety, DeepSeek-R1 shows the highest under-triage rate in this study (43.6%). On level-D (emergency) cases specifically, under-triage rates rise to 66.7% for DeepSeek-R1 and 58.3% for Mistral 7B, suggesting these models are particularly unreliable for the most critical presentations. ChatGPT Health has the lowest under-triage rate (5.1%) but the highest over-triage rate among the top performers (10.3%). Gemma 2 9B offers the most balanced safety profile among open-source models, with a 12.8% under-triage rate and 7.7% over-triage rate.

**Table 1: Comprehensive Evaluation of Model Performance and Safety across Clinical Triage tasks, N=78 cases**

| Model | Accuracy | Calibration | Coherence Rate | Faithfulness Rate | Under-Triage Rate | Over-Triage Rate |
|---|---|---|---|---|---|---|
| ChatGPT Health | **84.6% (4.1%)** | −2.7% | **52.6% (5.7%)** | 30.8% (5.2%) | **5.1% (2.5%)** | 10.3% (3.4%) |
| Gemma 2 9B | 79.5% (4.6%) | **−1.6%** | 35.9% (5.4%) | 43.6% (5.6%) | 12.8% (3.8%) | **7.7% (3.0%)** |
| Qwen2.5 7B | 73.1% (5.0%) | +8.2% | **52.6% (5.7%)** | 42.3% (5.6%) | 11.5% (3.6%) | 15.4% (4.1%) |
| Mistral 7B | 59.0% (5.6%) | +18.4% | 41.0% (5.6%) | 33.3% (5.3%) | 33.3% (5.3%) | 7.7% (3.0%) |
| DeepSeek-R1 7B | 50.0% (5.7%) | +31.2% | 34.6% (5.4%) | **55.1% (5.6%)** | 43.6% (5.6%) | 6.4% (2.8%) |

Values in parentheses are standard errors, $\sqrt{\hat{p}(1-\hat{p})/N}$, $N = 78$. Calibration (signed mean difference) is reported without SE.

## 4.3 Effect of Adding Clinical Data

Adding clinical data such as vitals, exam findings, and lab results to model inputs improved triage level assignment accuracy for four of five models, with gains ranging from +5.2% (Gemma 2) to +17.9% (DeepSeek) and +15.4% (ChatGPT), with Mistral being the exception, where accuracy declined (−10.2%). This pattern is consistent with Mistral 7B's broader sensitivity to input structure, discussed in the next subsection: the model's performance degrades when inputs deviate from a format its reasoning mechanism handles well, whether due to added clinical complexity or prompt structure changes. However, Faithfulness Rate degraded for all models when this additional clinical data was provided: the drop ranged from −38.5% for DeepSeek to −61.5% for Mistral. Notably, models elaborated on clinical data rather than grounding reasoning in it, generating reasoning that extends beyond the patient's stated information. ChatGPT Health followed the same pattern (Faithfulness Rate: 10.3% on full clinical prompts vs. 51.3% on symptoms-only prompts), indicating that this artifact is not limited to open source models. Emergency detection performance (i.e., in cases with gold standard labels of triage level D) for both ChatGPT and Gemma 2 worsened when labs were provided as inputs, suggesting that clinical context can perhaps confuse models causing them to miss the clearest urgency signals. Supplementary Table[1] shows the model sensitivity according to clinical content.

## 4.4 Prompt Format Sensitivity

To assess sensitivity of the models to prompt structure, we re-evaluated all open-source models using a stricter prompt that limits model reasoning to two to three sentences, keeping all clinical inputs brief. The effect of this change to the prompt varies substantially by model: Mistral 7B gained 15.4% in accuracy under the stricter prompt (59.0% to 74.4%), while DeepSeek-R1 7B lost 20.5% (50.0% to 29.5%). These diverging effects likely reflect architectural differences in how models use reasoning length: DeepSeek-R1 7B depend on extended chain-of-thought design that relies on reasoning space to reach conclusions; constraining it to two to three sentences removes the reasoning process the model depends on, while by contrast, Mistral 7B's free-form outputs introduce noise that stricter formatting suppresses. Gemma 2 9B and Qwen2.5 7B remain comparatively stable (−3.9% and −2.6% respectively), suggesting these architectures are less sensitive to prompt structure. Faithfulness to Clinical Context also shifts for Gemma 2 9B's where the Faithfulness Rate rises from 43.6% to 74.4% under the stricter

[1]https://github.com/sreerammarimuthu/AI-triage-benchmark

prompt, indicating that constraining reasoning length reduces fabrication even when accuracy is largely unchanged.

## 5 PROJECT VISION

This study establishes a reproducible multi-metric baseline for triage level assignment, and the results surface clear gaps we plan to address:

**New data expansion.** We applied the same four models to a confidential 21-case Emergency Department gold standard dataset annotated by two emergency physicians, evaluating 72 feature set compositions including physician-labeled Q&A pairs across 6,048 total predictions. The results were consistent with our main study; information quality, not volume, drove the performance, we plan to scale this with more annotated cases.

**Factorial stress-testing.** Ramaswamy et al. identified social anchoring biases that significantly shifted triage recommendations in edge cases. We will extend evaluation to all 16 factorial conditions, including demographic framing and barriers to care, measuring how hallucination and safety metrics shift across model families, and plan to introduce additional factorial conditions.

**Hallucination mitigation.** The accuracy-faithfulness tradeoff is the central unsolved problem, the most accurate model fabricates in 69.2% of its explanations. We will test retrieval-augmented generation and faithfulness-aware fine-tuning to close this gap.

**A better judge.** The current single-model judge cannot reliably capture all fabrications. We plan to improve judge prompts, explore stronger models, and evaluate an LLM council with adjudication for more reliable hallucination verdicts.

**Open benchmark release.** Once complete, we will release the full pipeline, vignette dataset with provenance metadata, judge prompts, and metrics module as a reusable open benchmark, directly addressing the reproducibility gap in clinical AI evaluation.

## 6 CONCLUSION

Consumer health AI is making triage decisions at scale without the evaluation infrastructure to match. This study shows that response accuracy alone is insufficient: the most accurate model (ChatGPT Health) fabricates clinical details in 69.2% of its explanations [10], prompt format shifts accuracy by up to 20.5%, and emergency-level cases are under-triaged at rates as high as 43.6%, none of which surface in a score that measures accuracy alone. Gemma 2 9B emerges as the most practical open-source option, matching ChatGPT Health within 5% on accuracy while offering a more balanced safety profile. These findings point to a clear need for multi-metric evaluation

infrastructure in clinical AI: reproducible, hallucination-aware, and safety-sensitive. The pipeline, benchmark, and roadmap presented here are a step toward establishing that standard.

## AUTHORS

**Sreeram Marimuthu** *(data management community)* is a researcher at the YOSSI Lab at WPI and a Data Scientist at Mass General Brigham. His work spans across human-in-the-loop systems, LLM evaluation, data management, machine learning and public health, with a current focus on developing AI systems that are more reliable, fair, and effectively applied in medicine & healthcare.

**Roee Shraga** *(data management community)* is an Assistant Professor of Computer Science and Data Science at WPI, leading YOSSI Lab. His research focuses on data discovery, integration, and versioning in complex environments such as data lakes, leveraging techniques from data management, machine learning, and human-in-the-loop systems. His work has appeared in top-tier venues including SIGMOD, VLDB, SIGIR, WWW, and ICDE, and has been recognized with awards such as the NSF CRII and BSF Start-Up.

**Xiaozhong Liu** *(biomedical & NLP community)* is a Professor of Computer Science and Data Science at Worcester Polytechnic Institute, where he leads research at the intersection of large language models, clinical AI, and personalized health monitoring. With over 150 peer-reviewed publications and 22 patents, his work spans AI-driven oncology, wearable-based digital twins, and privacy-preserving patient modeling. Prior to WPI, he served as an Associate Professor at Indiana University Bloomington.

**Patricia L. Mabry** *(biomedical community)* is a Research Investigator at HealthPartners Institute integrating simulation modeling, AI, and behavioral and social science to address health system challenges. She founded and directed the Systems Science and Health program at NIH OBSSR. Her current projects include AI-based clinical decision support and sovereign medical digital twins.

## ACKNOWLEDGMENTS

Shraga's work was supported in part by National Science Foundation (NSF) under award number IIS-2348121 and by the United States-Israel Binational Science Foundation (BSF).

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
