# OpenReview forum: "Right for the Wrong Reasons: A Benchmark for Hallucination and Clinical Safety in AI Health Triage"
_VLDB.org/2026/Workshop/BioDMS — BioDMS 2026 ProjectTalk_

### Official Review · Reviewer_XhE6 · 2026-06-01

**Summary:**

The paper presents an experimental study of four open source language models on an AI health triage task, together with a roadmap for a benchmark release.

**Confidence Of Review:**

3

**Detailed Feedback Points:**

S1. Quantifying the reliability and accuracy of AI given health advice is a problem with high real world importance.

S2. The experimental study is well designed with multiple metrics measuring different aspects of the problem, and has great potential for being turned into an open benchmark

W1. All evaluated models are relatively small, it would be interesting to see what happens with slightly larger models which still run consumer hardware via ollama. Furthermore, agentic tooling like web search could improve the results and is easy to integrate with existing models.

W2. Some experimental results are hard to explain and raise questions about the preliminary results in general, e.g., the extreme changes in result quality due to prompt changes or the negative effect of clinical data for mistral. Furthermore, it might have been helpful to use a stronger model than Llama-8B for the judge.

W3. The GitHub repository linked in the paper only contains the final results, but no code or data to reproduce them.

W4. The paper could make a stronger case, why the problem actually needs data management research (and could for example not be solved with NLP expertise alone).

**Relevance For Biodms:**

3

---

### Official Review · Reviewer_vyyV · 2026-06-08

**Summary:**

Dear Authors,

I have reviewed your manuscript entitled “Right for the Wrong Reasons: A Benchmark for Hallucination and Clinical Safety in AI Health Triage.” The authors introduces a benchmark for evaluating consumer-facing clinical triage systems, focusing on reasoning quality and patient safety. More precisely, they compare four open-source language models against a ChatGPT Health baseline using 78 physician-labeled clinical vignettes spanning 19 medical domains. In terms of evaluation metrics, they not only consider triage accuracy, but also calibration, reasoning coherence, faithfulness to the clinical context, and over- and under-triage rates through a reproducible two-stage evaluation pipeline. The results highlight trade-offs between accuracy and reasoning faithfulness and the impact of prompt design on model performance.

**Confidence Of Review:**

4

**Detailed Feedback Points:**

1) The paper focuses on an important problem, that is, evaluating clinical AI systems only based on their accuracy may overlook clinically relevant failure modes, and the proposed inclusion of calibration, reasoning coherence, faithfulness, and triage-related safety metrics provides a broader perspective on model performance.


2) The manuscript presents a reproducible evaluation framework and benchmark pipeline to conduct systematic comparison of multiple models.


3) A major limitation is the relatively small evaluation dataset. The benchmark is based on only 78 clinical vignettes derived from a previously published dataset. Given the small size of the datasets, some performance differences correspond to only a few cases, making it difficult to assess the robustness and generalizability of some of the conclusions.


4) The most significant concern relates to the use of a single LLM judge to assess coherence and faithfulness. Although the authors acknowledge this limitation in the “Project Vision” section, many of their conclusions rely on these metrics. Therefore, the manuscript would be significantly strengthened by additional validation of the judge’s reliability, for example through comparison with human annotation.

**Relevance For Biodms:**

3

---

### Official Review · Reviewer_KLhv · 2026-06-10

**Summary:**

This paper argues that accuracy alone is an inadequate metric for consumer-facing clinical-triage LLMs.
It introduces a benchmark that jointly evaluates accuracy, calibration, clinical reasoning coherence, faithfulness to clinical context, and under-/over-triage rates.
It compares four open-source models against a ChatGPT Health baseline via a two-phase, checkpoint-based pipeline with an LLM judge (Llama 3.1 8B).
The paper shows that ChatGPT Health is most accurate (84.6%) but fabricates clinical detail in 69.2% of explanations, while DeepSeek-R1 is least accurate but most faithful.

**Confidence Of Review:**

3

**Detailed Feedback Points:**

1. The distinction between being right and being right for the right reasons in safety-critical triage is important.
2. The pipeline based on Ollama makes it really easy for reproducing and testing more datasets.

Major concerns:
1. The main quantitative claims currently rest on an unvalidated judge. Llama 8B models is often not a good choice for judging subtleties and vagueness of produced responses. Validating the judge against physician labels and adding statistical rigor would make the high-level takeaways strong.
2. The data management angle is not completely clear. Not sure how it overlaps with the BioDMS community.

**Relevance For Biodms:**

1